# High-Throughput Volatilome Fingerprint Using PTR–ToF–MS Shows Species-Specific Patterns in *Mortierella* and Closely Related Genera

**DOI:** 10.3390/jof7010066

**Published:** 2021-01-19

**Authors:** Anusha Telagathoti, Maraike Probst, Iuliia Khomenko, Franco Biasioli, Ursula Peintner

**Affiliations:** 1Institute of Microbiology, University of Innsbruck, Technikerstrasse 25, 6020 Innsbruck, Austria; Ursula.Peintner@uibk.ac.at; 2Fondazione Edmund Mach, Research and Innovation Centre, Food Quality and Nutrition Department, Via Edmund Mach 1, 38010 San Michele all’Adige, Italy; Iuliia.Khomenko@fmach.it (I.K.); Franco.Biasioli@fmach.it (F.B.)

**Keywords:** volatilomes, *Podila*, *Linnemannia*, *Entomortierella*, Mortierellomycotina, PTR–ToF–MS, fungal volatiles

## Abstract

In ecology, Volatile Organic Compounds (VOCs) have a high bioactive and signaling potential. VOCs are not only metabolic products, but are also relevant in microbial cross talk and plant interaction. Here, we report the first large-scale VOC study of 13 different species of *Mortierella sensu lato (s.*
*l.*) isolated from a range of different alpine environments. Proton Transfer Reaction–Time-of-Flight Mass Spectrometry (PTR–ToF–MS) was applied for a rapid, high-throughput and non-invasive VOC fingerprinting of 72 *Mortierella s.*
*l.* isolates growing under standardized conditions. Overall, we detected 139 mass peaks in the headspaces of all 13 *Mortierella s.*
*l.* species studied here. Thus, *Mortierella*
*s.*
*l.* species generally produce a high number of different VOCs. *Mortierella* species could clearly be discriminated based on their volatilomes, even if only high-concentration mass peaks were considered. The volatilomes were partially phylogenetically conserved. There were no VOCs produced by only one species, but the relative concentrations of VOCs differed between species. From a univariate perspective, we detected mass peaks with distinctively high concentrations in single species. Here, we provide initial evidence that VOCs may provide a competitive advantage and modulate *Mortierella s.*
*l.* species distribution on a global scale.

## 1. Introduction

Microbial communication, interaction, and competition are mediated by a multitude of organic molecules produced by organisms in their given habitat [1]. Such molecules are regarded as secondary metabolites, many of which have been extensively studied to date. 

Due to their volatile nature, microbial Volatile Organic Compounds (VOCs) are predestined to facilitate the information exchange between microbes and their environment [2,3]. They might increase the competitive advantage of specific microorganisms, or trigger mutualistic interactions, thus enabling the colonization of new habitats, depending on nutrient availability and pre-colonization conditions [4,5]. Fungi produce different chemical classes of VOCs such as alcohols, aldehydes, ketones, esters, sesquiterpenes, and many others [4]. However, a knowledge gap exists regarding species-specific production and function of fungal VOCs. Greater insight into fungal volatilomes will facilitate a better understanding of fungal ecology. A systematic volatilome investigation based on a large number of species within a single genus is still missing.

Fungal species belonging to Mortierellaceae are known as important saprobic organisms living on a wide range of organic substrates like soil, plant debris, or animal dung [6,7,8]. Recent studies on the soil microbiota on a global scale reported *Mortierella* as a key player of the soil core microbiome [9,10] The phylogeny of Mortierellaceae has recently been resolved based on a multi-genome phylogeny and phylogenomics [10]. Thirteen genera are now recognized circumscribing *Mortierella sensu lato* (*s. l.*). *Mortierella sensu stricto (s. s.*) represents the *M. alpina* clade only, as it includes the type of the genus *M. polycephala*, as well as *M. alpina*, *M. globalpina*, and *M. antarctica*. Based on this modern Mortierellaceae taxonomy, our study includes species belonging to the genera *Mortierella (s. s.)*, *Podila* (*M. horticola*, *M. humilis*, *M. verticillata*), *Linnemannia* (*M. bainieri*, *M. exigua*, *M. gamsii*, *M. hyalina*, *M. solitaria*, *M. zonata*), and *Entomortierella* (*M. parvispora*). Several species have not been formally recombined in the new genera, and the correct affiliations of *M. angusta*, *M. pseudozygospora*, and *M. gemmifera* are still unknown. We, therefore, address these taxa with their *Mortierella* epithets.

Little is known about *Mortierella* ecology; however, there is increasing evidence that these fungi may have other functions beyond just being saprobic. *Mortierella globalpina* can be used as a potential biocontrol organism against nematodes and was shown to be associated with increased plant growth [11]. A plant-beneficial effect was also reported for the closely related species *M. alpina*, *M. antarctica*, and *M. verticillata* [12,13,14]. Few studies report the biotechnological application of *Mortierella* in the food industry for fatty acid production [15,16,17]. Other *Mortierella s. l.* species were studied for their secondary metabolite [16] and garlic-like odor production [6,18]. It appears that Mortierellaceae metabolism is diverse, and based on the abovecited literature, their metabolites fulfill a variety of functions. However, information on their specificity and function remain scarce. It would be extremely interesting, but challenging, to study the source and function of VOCs produced under natural conditions in the soil [19]. The knowledge of the VOC production of individual species in pure culture is an important prerequisite for understanding VOC production and function under natural conditions.

Although several analytical techniques are available for both detection and quantification of VOCs, the costs, limited resolution, time-consuming laborious sample preparations, and sensitivity towards specific compounds limit scientific research. Proton Transfer Reaction Time-of-Flight Mass Spectrometry (PTR–ToF–MS) is a modern technique overcoming many of these classical limitations. PTR–ToF–MS is a direct injection mass spectrometry tool allowing for the time-resolved, high-sensitivity, online monitoring of VOC concentrations. Most VOCs, namely the ones with a proton affinity higher than water, can be simultaneously detected and their concentration can be estimated. Although isomeric compounds cannot be separated by PTR–ToF–MS, sum formulas and total concentration and total concentrations can be inferred for most mass peaks. The method provides a rapid, non-invasive fingerprint of absolute VOC concentrations in the sample’s headspace down to the ppt range [20]. Owing to the robustness and high resolution, the method has been extensively used to study the volatilomes of fungi, yeasts, and fruits [21,22,23,24,25].

In the present study, we used PTR–ToF–MS to analyze the volatilomes of a large number of *Mortierella s. l.* strains and species. We tested if *Mortierella s. l.* volatilomes can be measured by PTR–ToF–MS and used for species discrimination. To our knowledge, this is the first study addressing species-specific volatilomes of *Mortierella s. l.* species and, more specifically, the first large-scale study (>10 species) on a fungal genus and its sister genera. In order to exclude potential effects of substrate or environmental factors, we studied the volatilomes of 72 pure culture strains from 13 different *Mortierella s. l.* species under standardized laboratory conditions. We based this approach on the hypothesis that fungal species can be separated based on their volatilome. With the combination of multivariate analysis and a thorough data mining approach, we provide tentative identification of compounds and discuss potential ecologically relevant features of selected VOCs.

## 2. Materials and Methods 

### 2.1. Isolation of Mortierella s. l. Strains 

All the *Mortierella s. l.* strains used in this study (Table 1, Appendix A) were isolated from different types of soil (subalpine, alpine forest soils, and barren ground of a glacier successional site) in Austria (Table 2) in 2019. Isolation was done directly from soil samples or buried mesh bags filled with sterile quartz sand. Sampling was carried out from snow-covered soil or soil during the vegetation period. For the cultivation, the soil was directly plated on potato dextrose agar (PDA) (26.5 g potato dextrose and 15 g agar in 1 L of deionized water) or L_CA_ media [26] with antibiotics. Briefly, L_CA_ media consisted of 0.2 g yeast extract, 1.0 g glucose, 2.0 g sodium nitrate, 1.0 g potassium dihydrogen phosphate, 0.2 g potassium chloride, 0.2 g magnesium sulfate heptahydrate, 15 g agar in 1 L distilled water. The following combination of antibiotics was used: streptomycin sulfate (0.1 g L^−1^) dissolved in water and tetracycline (0.05 g L^−1^) dissolved in ethanol. Both solutions were filtered through a 0.22 µm Millipore syringe prior to use. Plates were incubated at 10 °C. If necessary, additional transfer steps were carried out under the same conditions to obtain the pure cultures of *Mortierella.* Overall, 72 strains belonging to 13 different *Mortierella s. l.* species were used in the study (Table 1; for more information, see Appendix A).

### 2.2. Identification of Obtained Mortierella Isolates 

Molecular identification of the selected fungal isolates was performed using sequence data of the rDNA-ITS region. Colony PCR was performed using the mycelium from 7-day-old cultures with few modifications [32]. The rDNA-ITS region was amplified by using standard primers ITS1 and ITS4 [33]. Sanger sequencing was carried out by Microsynth AG (Balgach, Switzerland). 

Identification was based on BLAST searches followed by phylogenetic analysis in comparison with sequences from type strains and additional reference strains downloaded from GenBank. Sequence Alignment was carried out using MEGA X [34] and manually adjusted. For the ML analysis and the parsimony bootstrap search, all positions with less than 95% site coverage were eliminated, and there were a total of 320 positions in the final dataset. The best Maximum Likelihood (ML) model was the Tamura-Nei model+G, as detected by MEGA X. Initial tree(s) for the heuristic search were obtained applying Neighbor-Join and BioNJ algorithms to a matrix of pairwise distances estimated using the Maximum Composite Likelihood (MCL) approach and then selecting the topology with superior log likelihood value (+G, parameter = 0.1589). 

Bootstrap analyses (1000 replicates) were conducted by Subtree-Pruning-Regrafting (SPR) algorithm level three, in which the initial trees were obtained by the random addition of sequences (10 replicates). Additionally, branch robustness was tested with Bayesian Inference in MrBayes 3.2.6 [35]. GTR was used as a substitution model, and a gamma distribution of rate variation across sites was chosen. For prior probability settings, defaults were kept. For the Markov Chain Monte Carlo (MCMC) analyses, four chains were run for 10 million generations, with trees being sampled every 5000 generations. The analysis was stopped as the convergence diagnostic (average standard deviation of split frequencies) was below 0.05 after 10 million generations. From the 20,000 sampled trees (for each of the 2 runs), 25% were discarded as burn-in before summary statistics were calculated (using sump and sumt commands). Diagnostic plots as well as the convergence diagnostics EES (Estimated Sample Size; min ESS around 10 K) and PSRF (Potential Scale Reduction Factor; 1.000 for all parameters) indicated stationarity. Phylogenetic trees were drawn with Figtree 1.4.4 [36].

### 2.3. Cultivation of Mortierella Isolates for Volatilome Analysis

The *Mortierella s. l.* isolates were carefully selected based on the results of the phylogenetic analysis of rDNA-ITS sequences. For each terminal clade representing a species, five strains were used as biological replicates. One agar block of about 3 × 3 mm containing mycelium was used as inoculum for each vial. From each strain, three technical replicates were plated into 20 mL glass vials containing 5 mL of PDA slant agar media closed with a screw cap with a silicon/PTFE septum. Inoculated vials were incubated at 20 °C with a loosely attached screw to ensure air supply for 3 d in darkness. As controls, vials containing growth media only (without fungal inoculum) and empty vials were incubated at the same conditions and included in the volatilome analysis. The VOC measurement was carried out from the headspace of the vials by proton transfer reaction–time of flight–mass spectrometry (PTR–ToF–MS) on the fourth day of incubation.

### 2.4. Analysis of PTR–ToF–Mass Spectrometer

The measurements of fungal volatilomes were performed by direct injection of the headspace fungal pure cultures into a commercial PTR–ToF–MS 8000 apparatus (Ionicon Analytik GmbH, Innsbruck, Austria) coupled to an adapted autosampler (MPS Multipurpose Sampler, GERSTEL, Mühlheim an der Ruhr, Germany). The drift tube conditions were as follows: 110 °C drift tube temperature, 2.8 mbar drift pressure, 530 V drift voltage with the ion funnel on. An ion funnel was operated at the end of the drift tube to improve sensitivity [37]. This leads to an E/N ratio of about 140 Townsend (Td), with E corresponding to the electric field strength and N to the gas number density (1 Td = 10^−17^ Vcm^2^). The sampling time per channel of ToF acquisition was 0.1 ns, amounting to 350,000 channels for a mass spectrum ranging up to *m*/*z* = 400. The sample headspace was withdrawn through PTR–MS inlet with 40 sccm flow for 60 cycles resulting in an analysis time of 60 s/sample. Pure nitrogen was flushed continuously through the vial to prevent pressure drop. The data preprocessing was performed to the method described elsewhere [38]. The raw dataset contained 403 mass peaks. It was reduced to 139 peaks by applying noise and correlation coefficient thresholds. The first removed peaks were not significantly different from blank samples; the latter excluded peaks highly correlated with other peaks> (|r| > 0.99), which corresponds for the most part to isotopologues of monoisotopic masses [39]. Head-space concentrations of tentatively identified compounds were calculated according to the equations in Lindinger et al. [20] and given in ppbV (part per billion by volume). An average value of 2.0 × 10^−9^ cm^3^ s^−1^ for the reaction rate constant has been used. Therefore, the concentrations of all compounds are linearly related and can be considered absolute concentrations in arbitrary units.

### 2.5. Data Analysis

Data were visualized and analyzed statistically using R [40] and relevant packages therein [41,42,43]. The data set of all *Mortierella s. l.* specimens consisted of 139 mass peaks. To assess the reproducibility of volatilomes among technical replications and across biological representatives of a species, the Euclidean distance matrices between PTR–ToF–MS samples were investigated. Using nonparametric Wilcox test, we tested whether within-species differences were smaller compared to between-species differences. Using the same procedure, we also tested whether the technical replications of a biological species representative were closer in terms of volatilomes compared to the distances to other strains of this species. The analysis was supported by Permutational Multivariate Analysis of Variance Using Distance Matrices (Adonis) applying 999 permutations. Also using Adonis on the Euclidean distance matrix, we tested for significant differences between species, locations (sites), seasons (summer vs. winter), and isolation matrices (soil vs. mesh bag).

Adonis analysis revealed significant differences in volatilomes between species. In order to pinpoint mass peaks accounting for major differences between *Mortierella s. l.* species, we used univariate analysis. First, normality and homoscedasticity of univariate data and variance distribution were analyzed using Shapiro and Levene test, respectively. In addition, histograms were drawn. Usually, data were not normally distributed, and distributions were tailed. Consequently, both analyses of variance (ANOVA) and Kruskal test were applied to test for univariate differences between species and we focused on non-parametric test results. As within-species heterogeneity was high in some cases, we considered mass peaks, which were significantly increased in certain species by both parametric and non-parametric tests, for ecological interpretation. As the number of significant mass peaks was still very high (118/139 mass peaks), we asked which mass peaks were increased both generally in *Mortierella s. l.* species and certain *Mortierella s. l.* species. To answer this question, the mass peak concentrations for each PTR–ToF–MS measurement were divided by the mean mass peak concentration of each respective sample. In the resulting data set, peaks with a numeric value >1 are higher, and peaks <1 are lower than the average concentration. Only peaks above a ratio of one in at least all technical replications (*n* = 3) of one strain were selected for further investigation. The resulting data set of 56 mass peaks was further divided into mass peaks generally increased in *Mortierella s. l.* species and specifically increased in certain species. For generally increased VOCs, mass peak concentrations were above the average concentration in at least 60% of the measurements performed within each species. The other mass peaks were considered specifically increased within a certain species. Pairwise differences were analyzed using post hoc tests (Tukey and Dunn test, respectively). In post hoc tests, *p*-values were corrected for multiple comparisons using Bonferroni Holm correction. For all analyses, a confidence interval of 95% was applied.

For cluster analysis (dendrograms), the median distance between species’ representatives was calculated based on Euclidean distances between volatilomes and on phylogenetic ML analysis. The resulting median distance matrices were used for average clustering, which was found to be the most robust and representative method for our data. Dendrograms were visualized using R package dendextent [44].

## 3. Results

### 3.1. Origin and Identification of Isolated Mortierella Strains

We obtained 72 different *Mortierella* isolates from different alpine and subalpine regions in Austria, in different seasons, and from either mesh bag or soil (Table 1). The isolates clustered into 13 different clades of *Mortierella s. l.* species (Figure 1). ITS sequence similarity ≥ 99% and statistical support (Bayesian posterior probabilities/Parsimony Bootstrap support ≥ 90%) for terminal clades were used as criteria for species identification. *Mortierella alpina* and *M. globalpina* as well as *P. humilis* and *P*. *verticillata* could not be differentiated based on their sequences. Consequently, these species pairs were analyzed as joint clades *M. alpina/globalpina* and *P. humilis/verticillata*, respectively. The locations differed in habitat conditions, including soil properties (Table 2). However, most of the *Mortierella s. l.* species were isolated from several forest habitats. Location effects influenced the distribution of at least some species (Figure 2). *Mortierella alpina/globalpina* and *M. solitaria* (unpublished—in review) were exclusively isolated from alpine barren ground habitats.

### 3.2. The Vast Majority of Detected VOCs was Produced by All Mortierella Species

Using PTR–ToF–MS, 139 mass peaks were detected after filtering across all *Mortierella s. l.* isolates in the test. Over all mass peaks, the technical replication of the cultures were not identified as a significant factor explaining data set variance (p_Adonis_ = 0.097, R^2^_Adonis_ = 0.016 on a Euclidean distance matrix). Most of the mass peaks were associated to a sum formula and have been tentatively identified based on literature and the mVOC databases (Lemfrack et al. 2017) (http://bioinformatics.charite.de/mvoc/) (85 compounds, >60%). Except for H_2_S, all compounds were organic. The majority of chemically identified VOCs seemed to be alcohols (26) and (un-)saturated carbohydrates (47). In addition to C, H, and O, 13 VOCs seemed to contain S, N, or both. Their mass distribution was homogeneous in terms of both molecular mass and number of carbon atoms, although slightly shifted towards a higher number of mass peaks with lower molecular mass. The mean and median number of carbon atoms of detected VOCs was 6; the maximum was 15.

At least 134 mass peaks out of the 139 total were detected in all specimens. There were no peaks solely found in one species. Consequently, we did not observe any species-specific mass peaks that could directly be used as presence-absence indicators for species identification.

### 3.3. Mortierella Species Differ in Their Volatilomes

We detected species-specific volatilomes, which enabled discrimination of *Mortierella s. l.* species. The concentrations of 118 out of 139 mass peaks differed between species (p_Kruskal_ < 0.05). From a multivariate perspective, 56% of the overall variation between mass peak concentrations could be explained by specimen belonging to different isolates (p_Adonis_ = 0.001 on a Euclidean based distance matrix). The differences between *Mortierella s. l.* species accounted for 28% of the dataset’s variance (p_Adonis_ = 0.001).

### 3.4. Mortierella Phylogeny Corresponds to Volatilome

Volatilomes appeared to be phylogenetically conserved in the *Mortierella s. l.* species. In general, the pairwise distances calculated from VOC concentrations between strains belonging to one species were significantly smaller compared to the pairwise distances between species (Appendix A), which is crucial for differentiating *Mortierella* species by their volatilomes. However, there were a few exceptions, more specifically *E. parvispora* and *M. pseudozygospora* (Appendix A). For *M. angusta*, *L. gamsii*, *E. parvispora*, *M. solitaria* and *M. zonata*, the distances between technical replications were not significantly lower compared to the differences between biological representatives of these species (Appendix A).

Biological replications (strains) of a species explained a high portion of variance (56%) within the VOC dataset. This is in agreement with the phylogenetic heterogeneity between isolates and species (Figure 1). The relatively small fraction of variance explained by species (28%) is likely due to a combination of high within species heterogeneity observed in clades potentially representing a species complex, like in the *E. parvispora*, *M. globalpina/alpina* clades.

In ostensible contrast, there was no relationship between the pairwise phylogenetic distances of the isolates’ ITS2 sequences and the pairwise differences observed in mass peak concentrations (p_Mantel_ = 0.77 on 999 permutations). The high within species heterogeneity in terms of mass peak concentrations faced almost zero variation in phylogenetic distances, due to the phylogenetic distances being calculated from a conserved, relatively short marker gene. In addition, being an unrooted tree, the differences between species might not accurately reflect the phylogeny of the genus, especially in the context of the overall tree of life. Instead, we compared dendrogram topologies based on median distances of ITS sequences and volatilomes. Species were generally well-resolved in both topologies (Figure 3), supporting our hypothesis that *Mortierella s. l.* species differed in their volatilomes.

### 3.5. Discriminating Mortierella s. l. Species by Which VOCs?

Knowing that *Mortierella s. l.* species can be discriminated based on their volatilomes, we asked for the most discriminative mass peaks. The number of mass peaks with significant differences between species was quite high, and out of 139 mass peaks detected, 118 differed significantly between species (p_Kruskal_ < 0.05, Figure 4, Appendix A (Kruskal-Dunn), Appendix A (ANOVA-TukeyHSD)).

As the number of 118 mass peaks with different concentrations between species was too high to be discussed one by one, we identified peaks that seemed characteristic for their producing strain and analyzed their tentative attribution. In other words, for each technical replicate in PTR–ToF–MS measurement, we referred to the concentration of each mass peak measured in this sample to the mean concentration of mass peaks of this sample. Ratios higher than the mean peak concentration of a strain were above 1, ratios lower than the mean mass peak concentrations of a strain were below 1. A total number of 56 mass peaks had a ratio of >1 in all species. They were considered generally increased in a *Mortierella s. l.* species. These selected peaks still conserved the species-specific pattern observed in the overall dataset (R^2^ = 0.28, p_Adonis_ = 0.001). For 13 of those, more information could be extracted based on their respective PTR–ToF–MS signals (Table 3). Interestingly, all of those peaks also differed in their concentrations between the species, despite their generally increased concentrations compared to other peaks. Those peaks with a ratio > 1 in >50% of all technical replications of only one or a few species were considered as species-specific peaks (35 peaks). Out of those, 15 peaks with chemical annotation were found specific for some or few *Mortierella* species.

Based on the species-specific peaks, *L. exigua*, *M. pseudozygospora*, *M. angusta*, *M. bainieri*, *P. hyalina*, *M. gemmifera*, and the clade of *P. humilis/verticillata* could be clearly discriminated (Figure 4). Criteria for the selection were tentative chemical annotation of the peaks and their concentration.

### 3.6. Location Effects on Mortierella Volatilomes

In order to study if *Mortierella s. l.* species can be discriminated by their volatilomes, all strains were analyzed under standardized conditions. Consequently, habitat effects, such as the influence of pH or soil organic matter on volatilomes, cannot be implied from the data presented here, and they were not the aim of our study. It has to be emphasized here that despite differential amounts, all *Mortierella s. l.* species studied here produced the same VOCs, independent of the original habitat conditions of the strains. Still, we wanted to test whether the origin of stains influenced the volatilomes of *Mortierella*. While the most important environmental properties of the sampling sites are known (Table 2), the detailed habitat conditions, from which each single strain was isolated, were not focused here.

Using variance partitioning on the VOC data, the location, from which the strains were isolated, did not explain a significant portion of variance (4.4%). In the same manner, isolation conditions, i.e., season (summer vs. winter) and matrix (soil vs. meshbag), did not impact the variance present in the dataset of mass peak concentrations (R^2^ < 0.05). The fact that species were heterogeneously distributed across sampling sites makes it impossible to separate a possible origin effect from a species effect (Table 1, Figure 3). *M. solitaria*, *M. alpina/globalpina* and *P. horticola* were isolated exclusively from alpine sites. The biological representatives of *L. exigua*, *P. hyalina*, and *M. bainieri* were isolated from three comparable subalpine *Pinus cembra* forests (Praxmar, Patscherkofel, and Kühtai) (Table 1).

### 3.7. Mortierella and Their Species-Specifically Enriched VOCs

For both, mass peaks generally increased in *Mortierella s. l.*, and species-specific mass peaks, and the vast majority were alcohols (16/56).

Species-specific mass peaks cannot be defined based on being exclusively produced by a species, but by their significantly higher concentrations. Species-specific mass peaks are produced in 100- to 1000-fold higher concentrations in one species compared to the other species (Figure 5).

Interestingly, mass peaks with high concentrations across all *Mortierella s. l.* species investigated were smaller in mass compared to peaks with high concentrations in only certain *Mortierella s. l.* species in terms of both mass (median_general_ = 61.03, median_species-specific_ = 93.03, p_Wilcox_ = 0.001) and number of carbon atoms contained (median_general_ = 3, median_species-specific_ = 5, p_Wilcox_ = 0.004).

## 4. Discussion

### 4.1. PTR–ToF–MS as a High-Throughput Method for VOC Detection

In this study, we analyzed the volatilomes of 72 isolates representing 13 *Mortierella s. l.* species using PTR–ToF–MS. The volatilome analysis system by PTR–ToF–MS in combination with an efficient data-mining strategy is a very time- and cost-effective strategy for characterizing volatilomes. One advantage is that this is a high-throughput method for measuring VOC emission in real-time. Within the present work, our first aim was to test if PTR–ToF–MS is an appropriate method for analyzing volatilomes of *Mortierella s. l.* pure cultures. A thorough definition of fungal species-specific volatilomes is a prerequisite enabling future high-throughput real-time studies on the interaction of *Mortierella* spp. with associated bacteria or with plants.

Besides the abovementioned advantages, the drawback of PTR–ToF–MS compared to GC–MS is the identification of compounds with equal masses. This was already pointed out by Guo et al. (2020), who compared the VOC production of four *Trichoderma* spp. measured by PTR–ToF–MS and GC–MS. In PTR–ToF–MS, sesquiterpenes (isomeric terpenes of sum formula C_15_H_24_) were detected as one strong signal produced by two of the four *Trichoderma* species. GC–MS measurements in turn showed that the two involved *Trichoderma* spp. produced different species-specific sesquiterpenes [23].

Also, in *Trichoderma*, there were no species-specific VOCs produced, meaning that there was no compound produced by only one species, but specificities were concentration-dependent [23]. In fact, all mass peaks contained in our data set were detected across all *Mortierella s. l.* species. However, based on the concentrations of mass peaks, even grown under standardized conditions, *Mortierella* species could be separated based on their PTR–ToF–MS volatilomes. Whether there were differences between *Mortierella s. l.* species within mass peaks, that is differences in the production of VOCs with the same molecular mass, or if species produced similar VOCs remains to be investigated on a finer level. However, PTR–ToF–MS volatilomes appear to provide sufficient resolution for efficient discrimination between most included *Mortierella s. l.* species.

Thus, we conclude that PTR–ToF–MS provides an excellent platform for a systematic investigation of fungal volatilomes. The ground-breaking advantages of this method are its high-throughput opportunity, and the real-time and non-invasive monitoring of VOC production. Thus, volatilomes can be monitored under changing environmental conditions, during different fungal growth phases, or during microbial interaction.

An unambiguous identification of compounds produced by such a large number of *Mortierella s. l.* strains was not the primary aim of this study. We wanted to test for species-specific volatilomes. However, PTR–ToF–MS mass peaks can be tentatively identified based on VOC databases and data mining [22,23,34]. We identified a number of mass peaks that discriminated *Mortierella* species. Those can be used for a fast, cost-effective screening of new isolates and are interesting in *Mortierella* biology and ecology. The mass peaks with differential concentrations among *Mortierella s. l.* species can help in formulating hypotheses in terms of metabolic pathways active in species/isolates and suggest potential regulators of Mortierellaceae metabolism and signaling. The answer to such questions, however, is beyond the scope of this study. For further identification, PTR–ToF–MS can be combined with GC–MS for targeted and fast discovery of fungal VOCs of particular interest. However, for our aim, high accuracy of the mass peaks’ chemical identification was not needed, confirming PTR–ToF–MS as the method of choice. Our results pave the way for future studies, which might include chemical precision, and may give directions on what to expect and focus on in terms of *Mortierella* volatile compound production.

### 4.2. All Mortierella Species Produce a Variety of VOCs, But VOC Concentrations Are Species-Specific

With this extensive study, we show for the first time that, independent of selective environmental forces, *Mortierella s. l.* species generally produce a high number of different VOCs, including VOCs that have not been reported for *Mortierella* up to now. Their high chemical diversity is striking, and a possible biological function of these VOCs is under investigation.

In addition to the high diversity of *Mortierella* volatilomes, we regard it as crucial knowledge, that under standardized conditions, *Mortierella s. l.* species produced the same VOCs in species-specific concentrations. Based on our VOC analysis, we could clearly discriminate *Mortierella s. l.* species, but could not detect a genus-specific pattern in *Mortierella s. l.* Adding to the similar VOCs produced among species, volatilomes even appeared to be partially phylogenetically conserved among Mortierellaceae, supporting the hypothesis of evolutionary defined traits.

In our study, despite the standardized conditions, the volatilomes of the *Mortierella s. l.* species did not appear independent of the isolates’ habitat. However, despite the large number of specimens investigated here, we could not detect a clear dependency (Figure 3). We hypothesize that the volatilomes of *Mortierella* will be influenced by environmental factors, such as nutrient availability [45], temperature, pH, or even microbial composition of the habitat. Those, too, are certainly important questions for future studies. In this context, the finding that mass peaks with high concentrations across all *Mortierella s. l.* species investigated were significantly smaller in both mass and size compared to mass peaks with high concentrations in only certain *Mortierella s. l.* species gains relevance (Table 3, Figure 5). Bigger metabolites are usually produced in a regulated manner due to their costs. Thus, we assume that the production of large molecules is a consequence of species-specific environmental selection. In contrast, smaller VOC production might be more conserved as their production is less costly, increasing the odds of keeping its production as a genus-wide trait.

### 4.3. Potential Function of VOCs Produced by Mortierella Species

Acetaldehyde and different types of alcohols were generally high in relative concentration across all *Mortierella s. l.* strains (Table 3). These compounds’ reported functions vary among fungal species, but they certainly play a role in stress response and/or defense mechanisms [46].

More specifically, we detected an increased production of a mass peak tentatively identified (t.i.) as ethyl acetate (*m*/*z* 89.060, C_4_H_8_O_2_H^+^) in all our *Mortierella s. l.* species. Ethyl acetate is generally known as a substance with antibacterial properties against gram-positive bacteria [47] and was also reported to have a nematicidal effect [48,49].

Also, we detected high concentrations of a mass peak (*m*/*z* 87.045, C_4_H_6_O_2_H^+^) t.i. as diacetyl, especially in *M. pseudozygospora*. This was surprising because, until now, this highly bioactive VOC had been reported only from *Bacillus amyloliquefaciens*. It showed high antifungal activities, against fungal plant pathogens such as, e.g., *Botrytis cinerea*, *Monilinia fructicola*, *M. laxa*, *Penicillium italicum*, *P. digitatum*, and *P. expansum*, and anti-bacterial and bacteriostatic effects towards Gram-positive and Gram-negative bacteria [50].

The production of antibacterial, anti-fungal, and nematicidal VOCs provides a competitive advantage for *Mortierella s. l.* under in situ conditions. Such traits enable these fungi to defend themselves and to trigger various interaction processes with other organisms. Several *Mortierella s. l.* species have been reported as beneficial microbes due to their positive effect on plant health and performance [51]. The production of highly bioactive fungal VOCs could be an important factor triggering such interactions. Such VOCs could also provide an important competitive advantage, explaining the key role of *Mortierella s. l.* in the core soil microbiome, which might also contribute to their species-specific distribution across different habitats.

The hypothesis of *Mortierella s. l.* benefitting from bioactive VOCs is supported by several VOCs being characteristically high in certain species compared to others in the present study. The mass peak (*m*/*z* 107.050, C_7_H_6_OH^+^, Figure 5), t.i. as benzaldehyde, for example, were specifically increased in a few *Mortierella* species (*M. bainieri*, *M. pseudozygospora*). Benzaldehyde, which possesses insecticidal, antioxidant, and antimicrobial properties, was reported to be produced by the bacterium *Photorhabdus temparata* [52].

Moreover, also higher concentrations of methyl mercaptane (*m*/*z* 49.011, CH_4_SH^+^) were detected in *M. pseudozygospora*. This VOC is known to repel herbivores and parasites. It is known as an intermediate or end product in the metabolism of certain sulphur compounds [53,54].

In addition to VOCs with a potential bioactivity, our results suggest several other purposes for VOCs, underlining their relevance in nature and emphasizing their diversity. As an example, some volatile Sulphur derivatives were specifically detected in *M. pseudozygospora.* Volatile Sulphur compounds have been reported to be crucial during in vitro microbial cross talk: Sulphur assimilation is not only essential for growth and virulence in fungi, as shown for *Aspergillus fumigatus*, but it is known for an unprecedented regulatory cross-talk by heavily affecting iron homeostasis [55]. *Pseudomonas aeruginosa*-derived volatile Sulphur compounds are directly assimilated by *A. fumigatus*, where they promote growth and trigger a synergistic interaction [56]. A more thorough study on Sulphur-containing VOCs produced by our *Mortierella* species is planned in the future and is beyond the scope of this paper.

Especially higher concentrations of cyanide derivates (*m*/*z* 28.019, CH_2_N^+^ and *m*/*z* 74.007, C_2_H_3_NSH^+^—Methyl isothiocyanate) were detected in *P. hyalina*. Cyanide derivatives trigger sequestration of the metal ions in the rhizosphere and thus enhance the availability of nutrients for both, the involved microbes and the plants. These mechanisms could be particularly important in alpine soil types with low nutrient availability [57,58].

Isoprene (*m*/*z* 69.070, C_5_H_9_^+^) was detected in higher concentrations in *M. gemmifera*. Isoprene is the most abundantly produced biogenic volatile organic compound (VOC) on Earth, and it is highly reactive. Most isoprene is produced by plants, algae, but the relative contribution by microorganisms remains unknown. Isoprenoids deriving from isoprene are an important antioxidant substance alleviating oxidative stress [59]. Also, isoprene production could also be necessary for synergistic interactions, because selected groups of bacteria, especially Proteobacteria and Actinobacteria, can use isoprene as a carbon and energy source [60,61].

3,5-dimethylcatechol (*m*/*z* 137.061, C_8_H_8_O_2_H^+^) was produced in a higher concentration by *L. exigua.* In bacteria, this compound is involved in the degradation of methylated compounds via the catabolic pathway [62].

Thus, these results highlight the variety of potential bioactivities and functions of the VOCs specifically produced by single *Mortierella s. l.* species. All these compounds might have important functions for interaction and communication (quorum sensing, chemotaxis) in their environment. The ability of species to modulate their volatilome might be groundbreaking for their distribution. Further investigations focusing on certain specific VOCs are necessary to analyze and confirm their functions in detail. Moreover, it is not yet clear whether or not such VOCs are also produced under natural conditions. Future studies should also mimic the environment to understand how the *Mortierella* volatilome changes following changes in habitat conditions and associations to other microbes. These investigations should be coupled with monitoring the gene expressions to understand the metabolic/biosynthetic pathways involved in the production of the VOCs.

## Figures and Tables

**Figure 1 jof-07-00066-f001:**
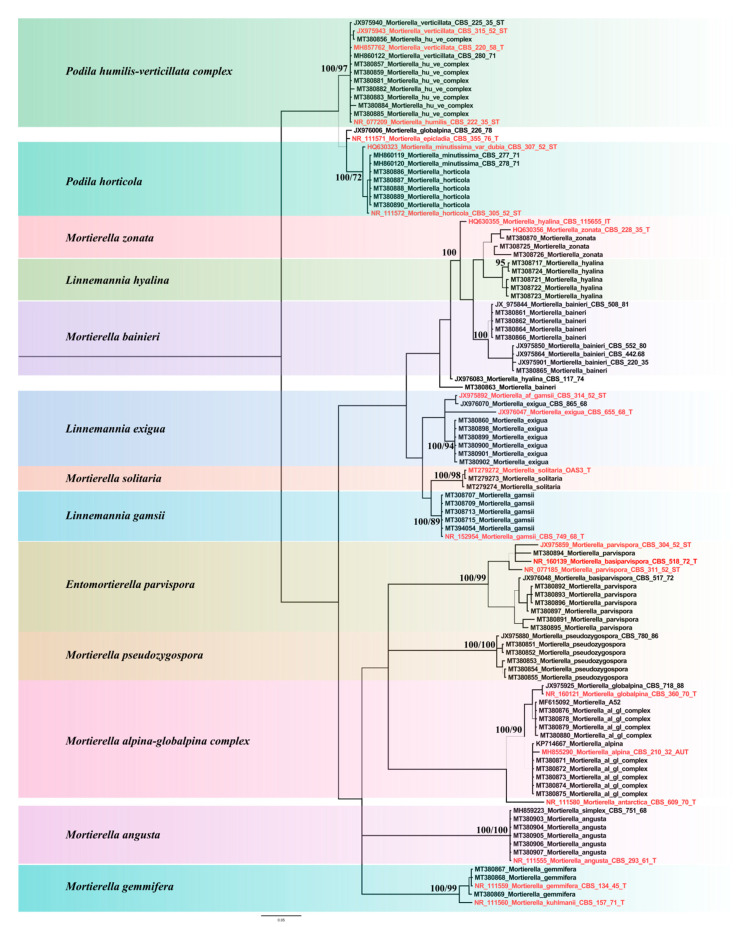
Phylogeny of the *Mortierella s. l.* species included in the VOC analysis. The 72 pure culture strains fall into 13 well-supported clades. Maximum Likelihood phylogram (log likelihood—1974.28) based on ITS sequences. Branch support (Bayesian posterior probabilities/Parsimony Bootstrap support ≥ 70) is shown above the respective branches. Sequences generated from type specimen are highlighted in red.

**Figure 2 jof-07-00066-f002:**
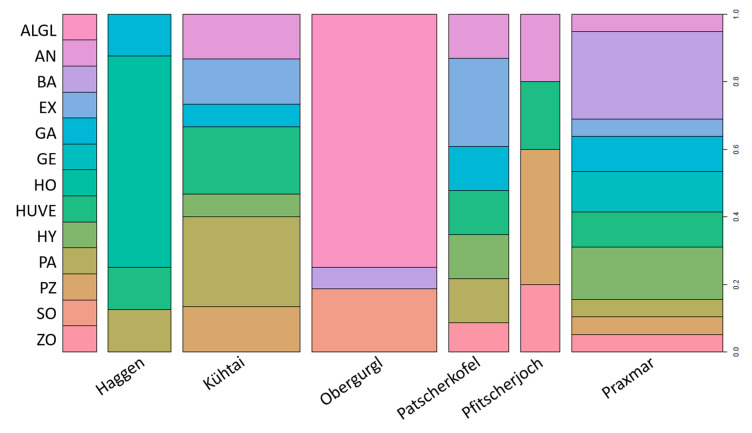
Overview of *Mortierella* s. l. isolates obtained from different locations. The percentages of different isolates (biological replicates) found in a certain sampling site are illustrated. The width of each bar corresponds to the relative sample size. Please note that the total numbers of isolates isolated from sampling sites cannot be inferred from the figure. ALGL = *M. alpina/globalpina*, AN = *M. angusta*, BA = *M. bainieri*, EX = *L. exigua*, GA = *L. gamsii*, GE = *M. gemmifera*, HO = *P. horticola*, HUVE = *P. humilis/verticillata*, HY = *L. hyalina*, PA = *E. parvispora*, PZ = *M. pseudozygospora*, SO = *M. solitaria*, ZO = *M. zonata*.

**Figure 3 jof-07-00066-f003:**
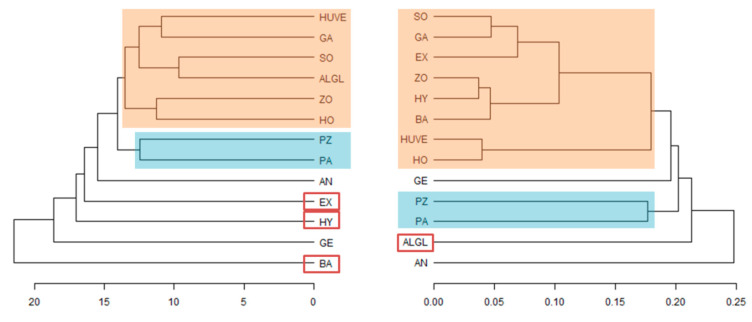
Dendrogram clusters of Mortierella *s. l.* species based on volatilomes (**left**) and ITS sequences (**right**). Clusters were calculated on an average cluster algorithm using median distances of biological replicates. ALGL = *M. alpina/globalpina*, AN = *M. angusta*, BA = *M. bainieri*, EX = *L. exigua*, GA = *L. gamsii*, GE = *M. gemmifera*, HO = *P. horticola*, HUVE = *P. humilis/verticillata*, HY = *L. hyalina*, PA = *E. parvispora*, PZ = *M. pseudozygospora*, SO = *M. solitaria*, ZO = *M. zonata*. Different colours (blue and pale orange) highlight identical clustering of species, the red rectangles show species with different clustering.

**Figure 4 jof-07-00066-f004:**
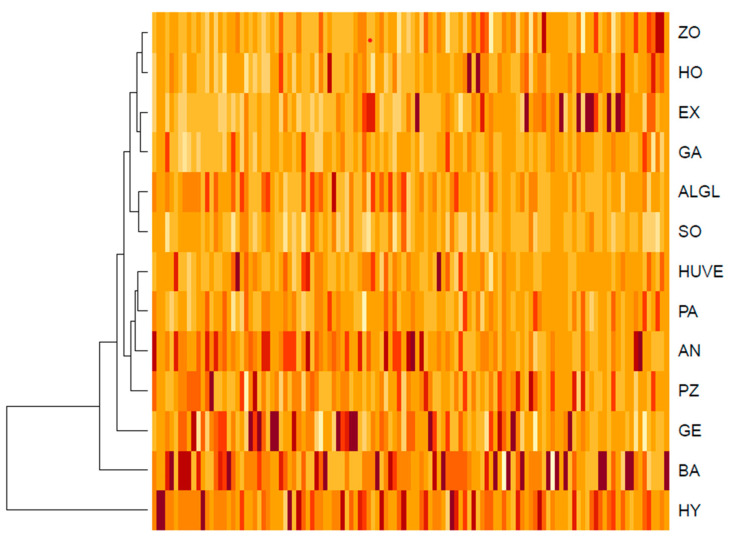
Relative concentrations of 118 mass peaks among 13 *Mortierella s. l.* species observed in this study. All mass peak concentrations illustrated differ significantly between species (p_Kruskal_ < 0.05). For illustration, the concentrations were centered and scaled by columns. Consequently, color shades can be compared between species but not between mass peaks. Darker colors indicate higher and lighter colors indicate lower concentrations. ALGL = *M. alpina/globalpina*, AN = *M. angusta*, BA = *M. bainieri*, EX = *L. exigua*, GA = *L. gamsii*, GE = *M. gemmifera*, HO = *P. horticola*, HUVE = *P. humilis/verticillata*, HY = *L. hyalina*, PA = *E. parvispora*, PZ = *M. pseudozygospora*, SO = *M. solitaria*, ZO = *M. zonata*.

**Figure 5 jof-07-00066-f005:**
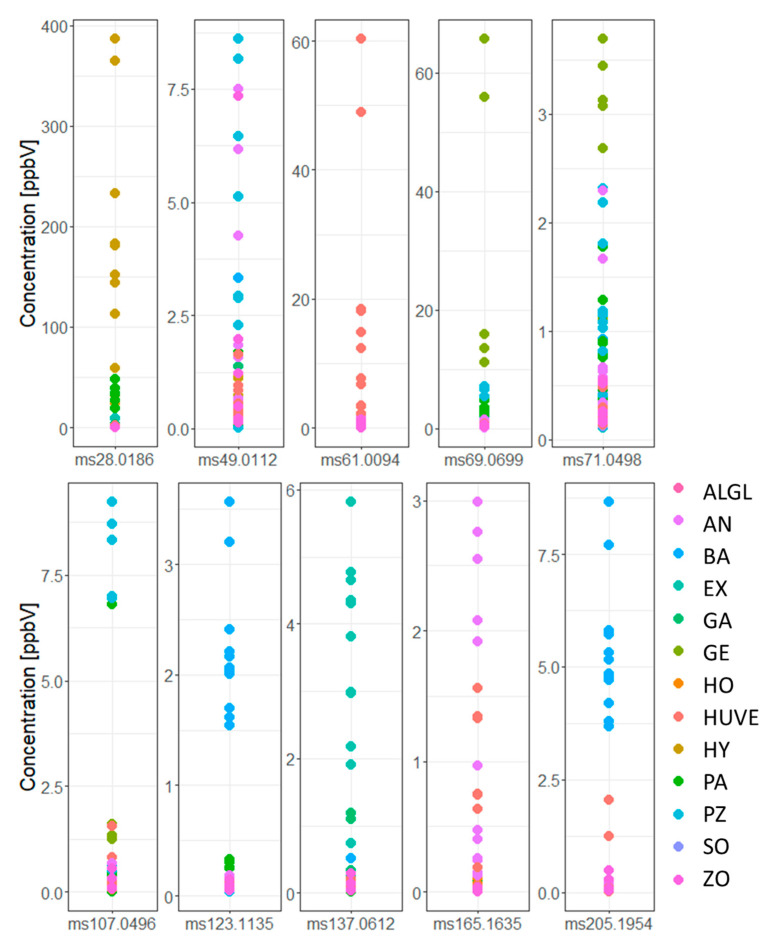
Mass peaks enriched in certain *Mortierella s. l.* species. Species differ significantly with regards to relative concentrations of these mass peaks according to Kruskal test and ANOVA (*p* < 0.05). ALGL = *M. alpina/globalpina*, AN = *M. angusta*, BA = *M. bainieri*, EX = *L. exigua*, GA = *L. gamsii*, GE = *M. gemmifera*, HO = *P. horticola*, HUVE = *P. humilis/verticillata*, HY = *P. hyalina*, PA = *E. parvispora*, PZ = *M. pseudozygospora*, SO = *M. solitaria*, ZO = *M. zonata*.

**Table 1 jof-07-00066-t001:** Origin of the analyzed pure culture isolates (*n* = 72) belonging to 13 different species of *Mortierella s. l.* (including *Entomortierella*, *Linnemannia*, *Mortierella s. s.*, and *Podila*) in alphabetical order based on species epithet. Information on sampling locations and conditions are provided; Season: snow-covered soil or soil during the vegetation period (=exposed); Matrix: isolated from buried mesh-bags filled with sterile quartz sand or directly from the soil.

Species	Location						Season		Matrix		Accession
	Haggen	Kühtai	Obergugl	Patscherkofel	Pfitscherjoch	Praxmar	Snowcover	Exposed	Meshbags	Soil	Numbers
*M. alpina/globalpina*	-	-	12	-	-	-	8	4	12	-	MT380876, MT380877, MT380878, MT380879, MT380880, KP714667, MF615092, MT380871, MT380872, MT380873, MT380874, MT380875
*M. angusta*	-	2	-	1	1	1	2	3	-	5	MT380903, MT380904, MT380905, MT380906, MT380907
*M. bainieri*	-	-	1	-	-	5	-	6	1	5	MT380861, MT380862, MT380863, MT380864, MT380865, MT380866
*L. exigua*	-	2	-	2	-	1	2	3	-	5	MT380898, MT380899, MT380900, MT380901, MT380860
*L. gamsii*	1	1	-	1	-	2	1	4	1	4	MT308707, MT308715, MT394054, MT308713, MT308709
*M. gemmifera*	-	-	-	-	-	2	-	2	-	2	MT380867, MT380868, MT380869
*P. horticola*	5	-	-	-	-	-	-	5	5	-	MT380886, MT380887, MT380888, MT380889, MT380890
*P. humilis/verticilata*	1	3	-	1	1	2	2	6	1	7	MT380856, MT380857, MT380859, MT380881, MT380882, MT380883, MT380884, MT380885
*L. hyalina*	-	1	-	1	-	3	1	4	-	5	MT308724, MT308721, MT308722, MT308717, MT308723
*E. parvispora*	1	4	-	1	-	1	5	2	1	6	MT380891, MT380892, MT380893, MT380894, MT380895, MT380896, MT380897
*M. pseudozygospora*	-	2	-	-	2	1	2	3	-	5	MT380851, MT380852, MT380853, MT380854, MT380855
*M. solitaria*	-	-	3	-	-	-	-	3	3	-	MT279272, MT279273, MT279274
*M. zonata*	-	-	-	1	1	1	2	1	-	3	MT308725, MT380870, MT308726

**Table 2 jof-07-00066-t002:** Description of sampling sites. The *Mortierella s. l.* strains were isolated from 10 sites in the Austrian Alps with distinct environmental conditions.

Location	pH	CN	Organic [% Dry Matter]	Coordinates	Altitude Masl	Dominating Trees	Citation
Kühtai 3	4	24.3	30	47.217208, 11.036823	2030	*Pinus cembra*	
Kühtai 2	3.5	19.9	70	47.208363, 11.006565	1880	*Pinus cembra*	
Kühtai 1	3	19.3	60	47.214527, 10.991395	1910	*Pinus cembra*	
Praxmar highest	3	21.1	10	47.155964, 11.128367	1820	*Pinus cembra*	[27]
Praxmar middle	3.5	14.2	30	47.154348, 11.130158	1780	*Pinus cembra, Picea abies*	
Praxmar lowest	4	13.9	50	47.162253, 11.139553	1520	*Pinus cembra, Picea abies*	
Haggen		20.6	25	47.21256, 11.087578	2230	*Pinus cembra* 35 years afforestation	[28]
Pfitschtal	4.5		35	46.994718, 11.668151	2261	*Salix retusa-reticulata* snowbed communities	
Obergurgl	7.5		0,6	46°50′ N, 11°01′ E	2400	Bare ground, glacier forefield	[29,30]
Patscherkofel	3.5	22.5	70	4.9938158, 11.6629124	2260	*Pinus cembra*	[31]

**Table 3 jof-07-00066-t003:** Mass peaks with generally high relative concentrations in *Mortierella s. l.* species. Values illustrate median and standard deviation of concentrations in [ppbV] across all technical replications of all biological representatives. The row “kruskal” indicates that the peak concentrations differ significantly between species (kruskal *p*-value). ALGL = *M. alpina/globalpina*, AN = *M. angusta*, BA = *M. bainieri*, EX = *L. exigua*, GA = *L. gamsii*, GE = *M. gemmifera*, HO = *P. horticola*, HUVE = *P. humilis/verticillata*, HY = *P. hyalina*, PA = *E. parvispora*, PZ = *M. pseudozygospora*, SO = *M. solitaria*, ZO = *M. zonata*.

	ms61.0295	ms73.0649	ms33.0339	ms48.0534	ms57.0404	ms60.0527	ms31.0183	ms43.0182	ms43.0545	ms41.0384	ms57.0702	ms39.0226	ms87.0445	ms89.0599
**Formula**	C_2_H_4_O_2_H^+^	C_4_H_8_OH^+^	CH_4_OH^+^	Isotope of C_2_H_6_OH^+^	C_3_H_4_OH^+^	Isotope of C_3_H_6_OH^+^	CH_2_OH^+^	C_2_H_3_O^+^	C_3_H_7_^+^	C_3_H_5_^+^	C_4_H_9_^+^	C_3_H_3_^+^	C_4_H_6_O_2_H^+^	C_4_H_8_O_2_H^+^
**Chemical**	Acid	Butanone	Methanol	Ethanol	General	Acetone	Form-	General	General	General	Butanol	General	Diacetyl	Ester
**Name**	Acetic acid	Butanal			fragment		aldehyde	fragment	fragment	fragment		fragment		Ethyl Acetate
**ALGL**	12 ± 5.6	11 ± 4.6	16 ± 4	2.2 ± 0.91	6 ± 0.74	8 ± 2.5	1.9 ± 0.54	9 ± 3.6	21 ± 4.8	15 ± 3.3	4 ± 5	19 ± 4.5	0.36 ± 0.069	5 ± 2.8
**AN**	8 ± 2.3	24 ± 6	12 ± 2.3	2.6 ± 0.89	6 ± 1.2	7 ± 1.5	2 ± 0.36	8 ± 4.1	24 ± 7.6	16 ± 4.9	4 ± 1.2	21 ± 6.5	0.5 ± 0.22	3 ± 2.1
**BA**	7 ± 5.3	20 ± 7.3	43 ± 16.4	1 ± 0.56	6.6 ± 0.71	8 ± 1.4	2.2 ± 0.28	6 ± 3.3	28 ± 4.6	21 ± 2.9	10 ± 2.4	28 ± 3.9	0.4 ± 0.32	2 ± 0.64
**EX**	6 ± 13.2	17 ± 5.7	17 ± 3.5	1.1 ± 0.48	5 ± 1.3	6 ± 2	1.7 ± 0.37	6 ± 7.2	17 ± 4.5	12 ± 3.1	2.9 ± 0.8	15 ± 4.3	0.5 ± 0.14	2.7 ± 0.99
**GA**	7 ± 16.5	16 ± 2.8	14 ± 5.5	1.2 ± 0.48	5.5 ± 0.64	8 ± 2.3	2.1 ± 0.49	6 ± 8.8	14 ± 3.6	11 ± 2.1	3.6 ± 0.77	15 ± 3.1	0.5 ± 0.34	2 ± 1
**GE**	9 ± 19.3	15 ± 6.5	16 ± 5.4	1 ± 1.1	6.6 ± 0.85	7 ± 1.4	2 ± 0.36	15 ± 8.1	11 ± 8.1	17 ± 5.2	3 ± 1.7	23 ± 10.6	0.7 ± 0.2	13 ± 4.2
**HO**	8 ± 8	13 ± 4.4	19 ± 4.1	0.8 ± 0.56	5.2 ± 0.72	7 ± 1.5	1.8 ± 0.29	7 ± 4.2	16 ± 3.8	12 ± 2.8	4 ± 1	17 ± 3.9	0.38 ± 0.096	2.1 ± 0.64
**HUVE**	8 ± 9.9	22 ± 7.4	15 ± 3.8	1 ± 0.42	6 ± 1.6	8 ± 2.2	1.9 ± 0.4	6 ± 4.4	18 ± 5.3	13 ± 3.5	3 ± 1.1	17 ± 4.8	0.4 ± 0.18	1.6 ± 0.71
**HY**	12 ± 6.4	9 ± 6	21 ± 9.7	1.8 ± 0.98	6.3 ± 0.92	7 ± 2.2	2 ± 0.43	10 ± 3.8	22 ± 6.9	16 ± 4.7	4 ± 1.3	21 ± 6.4	0.9 ± 0.6	6 ± 2.2
**PA**	7 ± 8.1	19 ± 10.5	10 ± 2.1	2 ± 72.2	6 ± 4.1	6 ± 2.8	2 ± 3.8	7 ± 15.5	19 ± 6.9	15 ± 18.5	4 ± 52.4	20 ± 30.4	0.5 ± 0.77	3 ± 5.1
**PZ**	11 ± 9.3	17 ± 5	13 ± 2.4	2 ± 1.5	6 ± 1.6	6 ± 1.7	1.8 ± 0.85	11 ± 5.3	23 ± 6.6	17 ± 4.9	3 ± 1.3	22 ± 8.4	1 ± 1.4	2 ± 3.3
**SO**	10 ± 3.1	6 ± 8.4	14 ± 2.1	1.5 ± 0.31	5.7 ± 0.67	6 ± 1.1	1.6 ± 0.27	8 ± 1.6	19 ± 5.4	15 ± 3.1	3 ± 0.79	20 ± 4.5	0.4 ± 0.24	5 ± 1.4
**ZO**	8 ± 41.9	13 ± 4	19 ± 9.3	0.9 ± 0.4	5.6 ± 0.59	7 ± 1.6	1.8 ± 0.29	6 ± 21.8	18 ± 4.5	13 ± 2.9	4 ± 1.1	17 ± 4.1	0.4 ± 0.16	2.1 ± 0.92
**kruskal**	9.3 × 10^−5^	2.1 × 10^−14^	2.3 × 10^−14^	5.7 × 10^−13^	4.8 × 10^−4^	1.0 × 10^−3^	9.7 × 10^−3^	1.2 × 10^−6^	5.4 × 10^−11^	1.2 × 10^−10^	2.8 × 10^−8^	4.1 × 10^−10^	1.4 × 10^−6^	6.9 × 10^−13^

## Data Availability

Please refer to suggested Data Availability Statements in section “MDPI Research Data Policies” at https://www.mdpi.com/ethics.

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
