# Peer review of "High-Throughput Volatilome Fingerprint Using PTR–ToF–MS Shows Species-Specific Patterns in Mortierella and Closely Related Genera"

_jof, 2021, doi:10.3390/jof7010066_

Round 1

Reviewer 1 Report

  • Check taxonomy with recently published by Gregory Bonito. Michigan State University
  • Double check species, names against the NCBI database, Taxonomy browser.
  • Potato Dextrose, Potato Dextrose Agar ??
  • Give full name of; LCA
  • Filtered antibiotic solution, Millipore filter.

Author Response

Check taxonomy with recently published by Gregory Bonito. Michigan State University

R: Thank you very much for this comment. We have integrated the work of Vandepol et al. 2020 into the revised version of our manuscript. (Please also see our response to the next comment in this regard.)

Double check species, names against the NCBI database, Taxonomy browser.

R: Thank you for this important comment. We double-checked and updated the taxonomy used in our work according to the NCBI taxonomy browser and to Vandepol et al. (2020). In cases of disagreement between the two, we used the taxonomy in Vandepol et al. (2020), which corresponds to Mycobank. As this taxonomy is very new and the taxonomy of Mortierellaceae cannot be considered fully resolved yet, we have added a paragraph in this regard into the introduction. Please see lines 46-55 in the revised version of the manuscript. As an overview, the following taxonomy has now been used:

New

Old

Comment

M. alpina/globalpina

M. alpina/globalpina

According to Vandepol et al. (2020) and NCBI.

M. angusta

M. angusta

According to Vandepol et al. (2020) and NCBI.

M. bainieri

M. baineri

We apologize for the typo. It has been corrected throughout the manuscript.

L. exigua

M. exigua

According to Vandepol et al. (2020).

L. gamsii

M. gamsii

According to Vandepol et al. (2020).

M. gemmifera

M. gemmifera

According to Vandepol et al. (2020).

P. horticola

M. horticola

According to Vandepol et al. (2020).

P. humilis/verticilata

M. humilis/verticilata

We apologize for the typo in one case (missing l in verticillata). In addition, we would like to mention that these species cannot be separated based on marker genes and, therefore, have been summarized to a complex here (which is in agreement with Vandepol et al. 2020).

L. hyalina

M. hyalina

According to Vandepol et al. (2020).

E. parvispora

M. parvispora

According to Vandepol et al. (2020).

M. pseudozygospora

M. pseudozygospora

According to NCBI.

M. solitaria

M. solitaria

We isolated and described this species. The description manuscript is currently still under revision.

M. zonata

M. zonata

According to NCBI.

Potato Dextrose, Potato Dextrose Agar ??

R: Yes, we are referring to potato dextrose agar (PDA). The abbreviation has been included into the methods section. Please see line 97 in the revised version of the manuscript.

Give full name of; LCA

R: The LCA medium does not come with a “full name”. Sometimes, it is also called Miura’s medium based on the original publication (in Japanese) in the 1970s (which is not publicly available). Therefore, we cited the Takashima et al. [28] who used exactly this medium and gave the full recipe in English. Based on our knowledge, his appears to be the common procedure for this medium’s recipe and preparation.

Filtered antibiotic solution, Millipore filter.

R: Yes, antibiotic solutions were sterile-filtered prior to use. The information has been added. Please see lines 102-103 in the revised version of the manuscript.

R: We addressed all comments raised in the .pdf in the list above.

Reviewer 2 Report

This is a fine paper, especially regarding the PTR methodology used to study the volatilomes of the fungi. its not the first dataset on this issue - a few other studies are also presenting data that are in line. However, the number of isolate/strains sampled are impressive and adds to the quality of the work. I'm not in doubt that volatilomics will play an increasing role in our understanding of microbial ecology. I have added my comments directly in the pdf of the manus. The are relatively few. One thing that should be improved, though, is the description of the conditions when VOCs were sampled. As far as I remember there is lack of information about: type of medium, size of sample vials, volume of headspace, age of mycelium, grown from spores or transferred as mycelium? Please give all these details. This is important as the growth conditions are heavily influencing the volatilomes of all kind of microorganisms. This is a well known fact from many studies as well as from own experience.

Reviewer 3 Report

Telagathoti et al., "High throughput volatilome fingerprint using PTR-ToF-MS shows species-specific patterns in Mortierella" descibes analyses of volatile organic compounds (VOCs) contained in the headspaces of lab-cultured Mortierella species. Very informative and a great use of this technology.

The manuscript is well-written and easy to read. 

Largest concern:

The concentrations of the compounds are not presented. For example, in Figure 5, there are no units for the concentration axes. If they are unknown and relative, use "relative abundance". Are those data linearly related for each compound? If not, are the statistical analyses valid?

Without knowing concentrations, the Discussion points regarding antimicrobial effects of ethyl acetate, diacetyl, and other VOCs are weakened because the reference studies evaluated known amounts of the test compounds.

Minor concerns:

A general tone throughout the manuscript that the detected VOCs should have a biological function is not warranted. Some may be simple metabolic pathway products or waste. Because each was grown on a non-natural substrate with abundant dextose, it is likely these microbes were in unique metabolic positions for this study. 

Line 331: space/hyphen needed "1000fold"

Author Response

Telagathoti et al., "High throughput volatilome fingerprint using PTR-ToF-MS shows species-specific patterns in Mortierella" descibes analyses of volatile organic compounds (VOCs) contained in the headspaces of lab-cultured Mortierella species. Very informative and a great use of this technology.

The manuscript is well-written and easy to read. 

Largest concern:

The concentrations of the compounds are not presented. For example, in Figure 5, there are no units for the concentration axes. If they are unknown and relative, use "relative abundance". Are those data linearly related for each compound? If not, are the statistical analyses valid?

R: We are grateful for this comment. We apologize that for some reason we missed indicating the calculation of concentrations and their respective units. The volatile concentrations were measured as absolute concentrations, though in arbitrary units. In the manuscript, the unit used is ppbV. The concentrations of all compounds are linearly related. We inserted the information into the methods section (please see lines 169-173) and adapted the text, figures and tables throughout the manuscript (e.g. Figure 5, Table 3).

Without knowing concentrations, the Discussion points regarding antimicrobial effects of ethyl acetate, diacetyl, and other VOCs are weakened because the reference studies evaluated known amounts of the test compounds.

R: As above, thank you for this valuable comment. Volatile compounds were measured in absolute concentrations and concentrations in ppbV are now indicated throughout the manuscript.

Minor concerns:

A general tone throughout the manuscript that the detected VOCs should have a biological function is not warranted. Some may be simple metabolic pathway products or waste. Because each was grown on a non-natural substrate with abundant dextose, it is likely these microbes were in unique metabolic positions for this study. 

R: This is true. We highlighted this issue specifically in the discussion. Please see lines 522-523 in the revised version of the manuscript.

Line 331: space/hyphen needed "1000fold"

R: Done.